# Greenhouse Gas Emission Offsets of Forest Residues for Bioenergy in Queensland, Australia

Leanda C. Garvie [1,*], Stephen H. Roxburgh [2] and Fabiano A. Ximenes [3]

1 Forest Research Institute, University of the Sunshine Coast, Sippy Downs, QLD 4556, Australia
2 Commonwealth Scientific and Industrial Research Organisation (CSIRO) Land and Water, GPO Box 1700, Canberra, ACT 2601, Australia; stephen.roxburgh@csiro.au
3 Forest Science, NSW Department of Primary Industries, Parramatta, NSW 2150, Australia; fabiano.ximenes@dpi.nsw.gov.au
* Correspondence: lgarvie@usc.edu.au; Tel.: +61-410-544-768

**Abstract:** Harnessing sustainably sourced forest biomass for renewable energy is well-established in some parts of the developed world. Forest-based bioenergy has the potential to offset carbon dioxide emissions from fossil fuels, thereby playing a role in climate change mitigation. Despite having an established commercial forestry industry, with large quantities of residue generated each year, there is limited use for forest biomass for renewable energy in Queensland, and Australia more broadly. The objective of this study was to identify the carbon dioxide mitigation potential of replacing fossil fuels with bioenergy generated from forest harvest residues harnessed from commercial plantations of *Pinus* species in southeast Queensland. An empirical-based full carbon accounting model (FullCAM) was used to simulate the accumulation of carbon in harvest residues. The results from the FullCAM modelling were further analysed to identify the energy substitution and greenhouse gas (GHG) emissions offsets of three bioenergy scenarios. The results of the analysis suggest that the greatest opportunity to avoid or offset emissions is achieved when combined heat and power using residue feedstocks replaces coal-fired electricity. The results of this study suggest that forest residue bioenergy is a viable alternative to traditional energy sources, offering substantive emission reductions, with the potential to contribute towards renewable energy and emission reduction targets in Queensland. The approach used in this case study will be valuable to other regions exploring bioenergy generation from forest or other biomass residues.

**Keywords:** forest residue; bioenergy; FullCAM; carbon dioxide emissions offsets; GHG emission offsets; climate change mitigation; fossil fuel substitution

## 1. Introduction

Greenhouse gas (GHG) emissions are the leading cause of rising global temperatures and anthropogenic climate change, and of these GHG emissions, carbon dioxide is the most significant contributor [1]. It is estimated that there has been a 48% increase in atmospheric carbon dioxide since the Industrial Revolution began over 250 years ago [1]. In 2021, the Intergovernmental Panel on Climate Change (IPCC) [2] reported on further evidence that emissions from human activities are responsible for increased global surface temperatures. Globally, energy in the form of electricity, heat, and transport accounts for over 70% of GHG emissions [3]. To fulfill the United Nation's ambition of limiting global warming to 1.5 degrees Celsius, there will need to be a 45% reduction in global carbon dioxide emissions from 2010 levels by 2030, reaching net zero around 2050 [4].

In Australia, the combustion of fuels for electricity and industrial processes accounted for 53% of total GHG emissions in 2020, with a further 17% of emissions coming from transport fuel combustion [5]. In Queensland, 13% of total GHG emissions can be attributed to transport [6]. Australia is a fossil-fuel-rich country that has experienced a highly politicized debate over several decades. However, commitments to international agreements, such as

the 2015 Paris Agreement where Australia committed to an emissions reduction target of 26% to 28% below 2005 levels by 2030 [7], and international conversations about reaching net zero emissions by 2050, have motivated the national debate about reducing fossil fuel combustion.

Bioenergy is energy generated from biomass. It is the oldest source of energy for humankind and is gaining renewed interest as nations seek to move away from fossil fuels [8]. Bioenergy generated from sustainably sourced feedstocks is expected to play a role in climate change mitigation in the coming decades [9]. In 2019, about 6.4% of Australia's energy consumption was from renewables, with about half of this being bioenergy [10], compared to 19.7% in Europe [11] and 11.3% globally (in 2018) [12]. Demand for bioenergy in Australia is expected to grow over the coming decades, contributing 20% or more of total electricity and transport fuels [13,14]. Initiatives to promote sustainable and renewable energies and emission reductions exist at both the national and state level. Under the Australian Government's Renewable Energy Target (RET) scheme, electricity providers are required to meet regulated renewable energy obligations [15]. The Emissions Reduction Fund (ERF) provides financial incentives for emission reduction technologies through the governmental purchase of carbon credits, which can also be traded [16]. The State of Queensland has made specific commitments to biofuels and bioproducts through its Biofutures 10-Year Roadmap and Action Plan, worth AUD 1 billion by 2026 [17], including the development of the AUD 16 million Advanced Biofuels Pilot Plant at Gladstone, Australia by Southern Oil Refining Pty Ltd., turning residues into aviation and other heavy industry fuels [17].

Bioenergy is versatile and can be produced from residues from forest or agriculture. It can be in the form of a solid, liquid, or gas, and can be transformed into heat, electricity, biofuels, and biogas [18]. Increasingly, industrial heat and power, or combined heat and power (CHP), is generated wholly or partially (co-fired with coal) from biomass feedstocks [19]. If biomass is used to generate grid-scale electricity in Queensland, it will most likely displace coal, because approximately 88% of the State's electricity is generated from coal [10]. Wood pellets are a source of heat and power across the world, including in Europe, Asia, and America. Wood pellets are a suitable substitute for gas boilers and heaters and, although there is some household use in Australia, the market is small, and most wood pellets produced there are exported to international markets. There are good opportunities for liquid biofuels to replace traditional diesels and have applications in transport, heavy industry, aviation, and shipping sectors in Australia and around the world [9]. When produced from lignocellulosic feedstocks such as wood residues, biofuels are generally referred to as renewable diesel, advanced biofuel, or second-generation biofuel, in contrast to traditional, first-generation biodiesels produced from vegetable oils and fats [20]. Renewable diesel is an established renewable energy that can be used with minimal or no modification to diesel engines. With large road networks and heavy reliance on road freight, there is significant scope for liquid biofuels to contribute to transport as well as heavy industry, shipping, and aviation fuel consumption in Australia.

Bioenergy can be generated from forest biomass which, in the developed world, includes harvest residues such as unmerchantable wood, wood removed during thinning, and other residue material. Forest residues are a sustainable source of bioenergy, involving the use of a by-product to replace fossil fuel alternatives, with the potential to lead to GHG emission savings [21]. Recent international studies have explored forest residue bioenergy and found several possible benefits, including emission reductions, and regional development opportunities in northern Spain [22], northern Italy [23], Canada [24], and the United States [25].

In Queensland, there are vast plantation forests covering approximately 230,000 hectares, over 85% of which are softwood plantations [26] generating large volumes of harvest residues, a potential feedstock for existing and future bioenergy projects. The annual estimate of forest harvest residues is up to 600,000 tonnes, and sawmill residues are estimated to be over 900,000 tonnes, mostly from softwood plantations with only minimal

contributions from hardwood native forests [27]. Plantation harvesting and thinning operations generate residues, some of which do not have a commercial use. During softwood plantation timber harvesting, only merchantable wood is extracted and removed from the plantation [28], leaving other fractions of the tree such as tree stumps, small branches, bark, and needles behind, as well as whole small trees and stems that are not suited to wood products [28,29]. Traditionally, forest harvest residues are left to decay on-site or burned to reduce the risk of forest fires. Increasingly, harvest residues are being captured for co-products such as wood chips, panel construction, landscaping, and bioenergy.

This study investigated the GHG mitigation potential of using forest harvest residues for bioenergy at a regional scale in Queensland, Australia. This research builds on previous biomass-for-bioenergy studies in Australia. Past research has estimated Australia's potential supply of bioenergy and GHG mitigation at a (macro) national scale [30]; [and the potential availability of a range of biomass types, including plantation harvest residues, on a broad, spatial level [31]. Moreover, the potential availability and costs of agricultural and forestry residues for electricity generation, replacing coal-fired electricity, in the Green Triangle region of southern Australia has been investigated [32], and in Tasmania, a previous study assessed native forest residues and the availability for bioenergy in Tasmania [33]. For the State of Queensland, other studies have provided information about the supply of lignocellulosic (dry plant biomass) feedstocks, specifically, grasses, short rotation trees, native vegetation regrowth for aviation fuels [34], and native plantation forest regrowth for aviation fuels in Queensland [35]. Other researchers have investigated sustainably managed private native forests [36] and private hardwood plantations [37] as a potential source of biomass for bioenergy in southeast Queensland. Other studies have investigated optimal locations for bioenergy facilities in Queensland, considering factors such as the availability of forest biomass [38] and multiple biomass feedstocks [39]; the economic and environmental viability of supplying bioenergy plants [40] have also previously been investigated for Queensland.

Few dedicated studies have quantified the implications of managed forest harvest residues for bioenergy from a climate change mitigation perspective on a regional scale. This study aims to obtain a better understanding of the GHG emission offsets of forest harvest residues for bioenergy in Queensland, Australia. Understanding biomass availability and net carbon implications, based on broad assumptions of forest practices and technologies, is an important first step and will be demonstrated here with a large, privately managed forest operation in southeast Queensland as a case study. In this study, carbon stocks and flows were estimated using the Full Carbon Accounting Model (FullCAM). FullCAM is an industry standard model used by the Australian Government to report carbon emissions on Australia's National Greenhouse Gas Inventory and to the United Nations Framework and Convention on Climate Change [41]. As well as application at the national scale for recording land sector GHG emissions, FullCAM is applied at the local scale for monitoring and reporting carbon sequestration projects under the Australian Government's Emissions Reduction Fund [16].

## 2. Materials and Methods

### 2.1. Study Area

The area of analysis was a softwood plantation managed for timber production, Toolara–Tuan Forest Estate, located in the Gympie–Maryborough region, roughly 200 km north of Brisbane, Queensland (25.99586 S, 152.83367 E) (Figure 1). It is a rural region that extends eastward to the coast, where it meets the Great Sandy Strait. Toolara–Tuan Forest Estate is one of several plantation estates throughout the state of Queensland managed by a single private company, Hancock Queensland Plantations Pty Ltd. (HQPlantations) (Gympie, Australia). The area is characterized by mild winters and warm, humid summers, with long-term average temperatures between 10 °C in July and 29 °C in January. The area receives high rainfall, with an annual long-term average of 1400 mm.

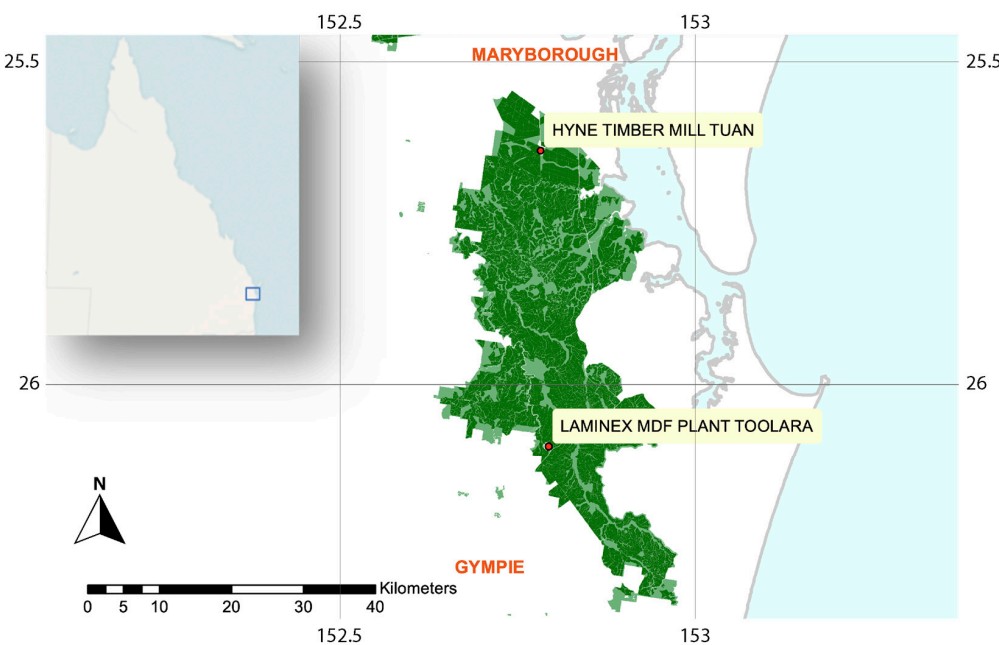

**Figure 1.** Map showing the location of Toolara–Tuan Forest Estate.

HQPlantations operates sustainable forest management practices according to certification standards of the Forest Stewardship Council (FSC) and Australian Forestry Standard for Sustainable Forest Management (AFS) [42]. The Toolara–Tuan Forest Estate is a long-rotation plantation with about 85,000 hectares of non-native, commercial, softwood *Pinus* spp., commonly referred to as Southern Pine. Of the three Pinus taxa (*Pinus caribaea* var*. hondurensis* [PCH], *Pinus elliottii* var*. elliottii* [PEE], hybrid pine [PEE x PCH]) in cultivation in the estate, the current profile is dominated by the locally developed hybrid pine [PEE x PCH].

## 2.2. Plantation Management Practices

The Toolara–Tuan Forest Estate is managed on 28–30-year rotation cycles. About 3% of the 85,000 hectares undergoes clearfell harvest each year [43], mostly through whole tree (WT) harvesting technique and is replanted following a site preparation phase. In WT harvesting, most of the above-ground portion of the trees is removed from the harvest site, whereas in cut-to-length (CTL) systems, stems are cut into shorter lengths at the harvest site. The silviculture regime typically involves site preparation, aerial herbicide application, hand planting in rows or mounds, fertilizer application as required, and chemical and mechanical weed control. Plantation management practices include thinning operations at around 14 years after planting, when approximately 40% of stems are removed.

Under current management (business as usual), stems removed during thinning operations are transported to wood processing operations located adjacent to the estate. A small proportion of tops (stems), branches and bark remain in situ to decay. Similarly, following clearfell harvest, small tree fraction proportions (5% of stems, plus branches and bark) remain in the field, while the bulk of the stems, high quality sawlogs and ply logs, are transported to a wood processing mill (Hyne Timber & Sons, Maryborough, Australia) co-located in the Toolara–Tuan Forest Estate. Traditionally, the bulky residue left behind at clearfell harvest is raked into windrows and either burned or left to decay, whereas smaller materials decay in situ. At times, when burning does not occur, the residue may instead be mechanically mulched in the field and spread out to decay. More recently, stem and branch residues from clearfell harvest are collected and chipped by a mobile chipper in the field and transported to co-located facilities as wood chips for bioenergy generation near the plantation estate.

### 2.3. Application of FullCAM to Estimate Carbon in Residues

Estimates of carbon stocks and flows of available harvest residues were not available from the plantation manager; therefore, they were made using FullCAM [44]). FullCAM is a freely available software system for tracking GHG emissions and changes in carbon stocks associated with land use and management in Australian agricultural and forest systems [41]. The model was configured to incorporate plantation-specific parameters, including the geographic location, *Pinus* spp., rotation length and timing, and management treatments such as thinning events (refer to Supplementary Figure S1 and Table S1). For this study, two FullCAM simulations were used to provide estimates of available residues over the course of a single rotation, corresponding to residue utilization alternatives 1 and 2 (Table 1). A single location, representing average climatic and soil conditions across the Toolara–Tuan Forest Estate, was selected to provide a representative estimate of forest residue production.

**Table 1.** Summary of bioenergy type, fossil fuel substitution, and tree fraction utilization of the different scenarios and residue utilization alternatives.

| Scenario | Bioenergy Product Type | Fossil Fuel Substitution | Residue Alternative | Forest Treatment | % Utilization | | |
|---|---|---|---|---|---|---|---|
| | | | | | Stem | Branch | Bark |
| 1 | CHP | Coal-fired electricity | 1 | Thin<br>Final harvest | 5 | 95 | 5 |
| | | | 2 | Thin<br>Final harvest | 95<br>5 | 95 | 95<br>5 |
| 2 | Pellets | Natural gas | 1 | Thin<br>Final harvest | 5 | 95 | 5 |
| | | | 2 | Thin<br>Final harvest | 95<br>5 | 95 | 95<br>5 |
| 3 | Renewable diesel | Diesel | 1 | Thin<br>Final harvest | 5 | 95 | 5 |
| | | | 2 | Thin<br>Final harvest | 95<br>5 | 95 | 95<br>5 |

### 2.4. Case Study Scenarios and Residue Utilization Alternatives

Avoided emissions were estimated for three different bioenergy scenarios with two residue utilization alternatives (Table 1). The scenarios varied according to the final bioenergy product type and the fossil fuel product that it could feasibly replace. Scenario 1 closely resembles business as usual (BAU), given that residues are currently utilized for on-site heat and energy by facilities co-located at the Toolara-Tuan Forest Estate (Laminex Group Pty Ltd., Maryborough, Australia); however in the current business model residues other than harvest residues are utilized. Scenario 2 also resembles an existing BAU, although currently, most of the feedstock used is sawmill residue. Co-located at the study site is Altus Renewables Limited (Maryborough, Australia), producing wood pellets from residues; however, in the current business model, the wood pellets are exported to international markets [45], and therefore carbon mitigation and renewable energy is attributed external to Queensland. Scenario 3 is a theoretical residue bioenergy option in Queensland; however, investment in and development of an advanced biofuels pilot plant by Southern Oil Pty Ltd. at Gladstone, Australia, approximately 300 km north of the study site, is expected to lead to demand for harvest residue in the future [46].

In Scenario 1, emissions avoided by replacing centralized State grid electricity with wood-fuelled integrated cogeneration, often referred to as combined heat and power (CHP), were estimated. CHP was assumed to mitigate 88% of emissions in this study, given that 12% of Queensland electricity is generated from renewable sources. In Scenario 2, emissions avoided by replacing natural gas with wood pellets were calculated, and in Scenario 3,

emissions avoided by replacing diesel fuel with renewable diesel for transportation and other industrial uses were calculated.

For each scenario, two residue utilization alternatives were investigated. These alternatives varied according to the volume of thinning material available for bioenergy. For softwood plantations, debarking does not occur in the field and the bark is generally harder to separate from the stems and branches. In this study, bark was assumed to be included in the residue at the same proportion as the stems for both thinning and clearfell operations. Residue utilization alternative 1 assumes that just a small proportion (5%) of thinning stems and bark are available for bioenergy, whereas alternative 2 assumes that whole stems (95%) and bark from thinning are available. For all scenarios, 95% of branches from thinning operations and final harvest were assumed to be available for bioenergy, as well as 5% of stems and bark from final harvest, which, in a commercial setting, includes stems outside standard width and shape dimensions. In this study, stumps, leaves and cones were not included in residue estimates and assumed to be left on-site.

### 2.5. Estimating Energy Potential and GHG Emissions Avoided

Carbon dioxide emissions avoided by substituting fossil fuels with residue bioenergy can be estimated by calculating the fossil-fuel-based emissions associated with generating an equivalent amount of energy as the bioenergy alternative. Net GHG emissions avoided in this study considered emissions displaced by combusting the substituted fossil fuel type, non-carbon dioxide emissions (methane and nitrous oxide) generated by combusting the bioenergy type (which are not avoided), and estimates of transport emissions associated with hauling residue to the bioenergy facility. Emissions associated with forest residue extraction and storage, and alternative residue treatments, such as field burning or decay, were beyond the scope of the current study. Net GHG emissions avoided, expressed as carbon dioxide equivalents, is the difference between fossil fuel type of carbon dioxide combustion emissions, and the bioenergy of non-carbon dioxide and transport carbon dioxide emissions. The bioenergy non-carbon dioxide emissions are methane and nitrous oxide emissions generated by the bioenergy type, estimated and expressed as carbon dioxide equivalents. Transport emissions are an estimate of carbon dioxide emissions associated with hauling harvest residue from felling site to bioenergy facility. Total net emissions avoided, therefore, are effectively GHG emissions (expressed as carbon emission equivalents) avoided. Table 2 summarizes the key parameters used in the calculations.

**Table 2.** Summary of main parameters used in the calculation of emissions avoided (Equations (1)–(14)).

| Parameter | Units | Values | | |
|---|---|---|---|---|
| **Harvest Residue** | | | | |
| Total available energy content in dry wood ($E_{content(wood)}$) | GJ t$_{biomass}^{-1}$ | 18.63 | | |
| Carbon content ($C_{content}$) | % | 50 | | |
| Total available energy content in carbon | GJ tC$^{-1}$ | 37.26 | | |
| **Renewable diesel** | | | | |
| Total available energy content ($E_{content(biofuel)}$) | GJ kL$^{-1}$ | 34.60 | | |
| Renewable diesel yield ($V_{prod}$) | kL t$_{biomass}^{-1}$ | 0.36 | | |
| **Scenarios** | | CHP | Pellets | Biofuel |
| Conversion efficiencies ($CF$) | % | 70.00 | 75.00 | - |
| Net available (useful) energy content ($E_{avail}$) | GJ tC$^{-1}$ | 26.08 | 27.95 | 24.80 |
| Product emission factors ($EF$) | kgCO$_2$-e GJ$^{-1}$ | 225 [1] | 51.40 | 69.90 |
| Non-CO$_2$ emission factors ($EF_{non-CO2}$) | kgCO$_2$-e GJ$^{-1}$ | 1.20 | 1.20 | 2.50 |
| **Transport emissions** | | | | |
| Diesel energy content ($E_{content(diesel)}$) | GJ kL$^{-1}$ | 38.6 | | |
| Diesel fuel intensity ($Fuel_{intens}$) | L tC$^{-1}$ km$^{-1}$ | 0.16 | | |
| Transport distance | km | 50 | 50 | 300 |

[1] Scope 2 emission factor.

### 2.5.1. Net GHG Emissions Avoided ($CO_2$-$e_{avoid}$)

Net GHG emissions avoided were calculated with:

$$CO_2 - e_{avoid} = CO_2 - e_{fossil} - (CO_2 - e_{gen} + CO_2 - e_{transp}) \quad (1)$$

where $CO_2$-$e_{avoid}$ is the net GHG emissions avoided, expressed as carbon dioxide equivalents, $CO_2$-$e_{fossil}$ is the carbon dioxide emissions associated with combusting the substituted fossil fuel type, $CO_2$-$e_{gen}$ is the methane and nitrous oxide emissions associated with combusting the bioenergy type, expressed as carbon dioxide equivalents, and $CO_2$-$e_{transp}$ is the carbon dioxide emissions associated with transporting residue biomass to the bioenergy facility.

### 2.5.2. CHP and Pellets ($CO_2$-$e_{fossil}$ and $CO_2$-$e_{gen}$)

To determine carbon dioxide emissions displaced and non-carbon dioxide emissions generated for solid fuels (CHP and pellets), it was necessary to first calculate the useful (available) energy contained within the carbon in the harvest residue. Assuming a moisture content of 15% and energy content of dry wood of 16.2 GJ tonne$^{-1}$ [47], the energy content of the dry wood biomass was estimated to be 18.63 GJ tonne$^{-1}$. This is consistent with the 19 GJ tonne$^{-1}$ reported previously [47,48].

Using the standardized value of 50% carbon content per weight in wood, and an assumed (base) energy conversion efficiency of 70% for CHP [49] and 75% for wood pellets [50], the energy per tonne of carbon for solid fuels was calculated with:

$$E_{avail(CHP)} = \frac{E_{content(wood)}}{C_{content}/100} \times \left(CF_{pellet}/100\right) \quad (2)$$

$$E_{avail(pellet)} = \frac{E_{content(wood)}}{C_{content}/100} \times (CF_{CHP}/100) \quad (3)$$

where $E_{avail(CHP)}$ and $E_{avail(pellet)}$ are the useful (available) energy (GJ tC$^{-1}$) in CHP and pellets, respectively, $E_{content(wood)}$ is the energy content of dry forest residue biomass (GJ t$_{biomass}$$^{-1}$), $C_{content}$ is the carbon content (%), and $CF_{CHP}$ and $CF_{pellet}$ are energy conversion efficiency factors (%).

Using available energy figures and the relevant NGA [47] emission factors (scope 2 electricity; scope 1 natural gas) of the substituted fossil fuel type, carbon dioxide emissions avoided through combustion were calculated with:

$$CO_2 - e_{fossil(CHP)} = E_{avail(CHP)} \times (EF_{CHP}/1000) \times 0.88 \quad (4)$$

$$CO_2 - e_{fossil(pellet)} = E_{avail(pellet)} \times \left(EF_{pellet}/1000\right) \quad (5)$$

where $CO_2$-$e_{fossil(CHP)}$ and $CO_2$-$e_{fossil(pellet)}$ are the carbon dioxide emissions associated with combusting the substituted fossil fuel type (tCO$_2$-e tC$^{-1}$), $E_{avail(CHP)}$ and $E_{avail(pellet)}$ are the useful (available) energy (GJ tC$^{-1}$) for each product type, $EF_{CHP}$ and $EF_{pellet}$ are emission factors for the production of either CHP or pellets using fossil fuel (kgCO$_2$-e GJ$^{-1}$), and 0.88 is the fossil fuel proportion in the State electricity grid [10].

Using available energy figures and the NGA [47] emission factors for methane and nitrous oxide generated through combusting dry wood, the non-carbon dioxide emissions generated in the bioenergy type were calculated with:

$$CO_2 - e_{gen(CHP)} = E_{avail(CHP)} \times \frac{EF_{non-CO_2(CHP)}}{1000} \quad (6)$$

$$CO_2 - e_{gen(pellet)} = E_{avail(pellet)} \times \frac{EF_{non-CO_2(pellet)}}{1000} \quad (7)$$

where $E_{avail(CHP)}$ and $E_{avail(pellet)}$ are as above, and $EF_{non-CO_2(CHP)}$ and $EF_{non-CO_2(pellet)}$ are the non-carbon dioxide emission factors for CHP and wood pellets (kgCO$_2$-e GJ$^{-1}$).

### 2.5.3. Renewable Diesel ($CO_2$-$e_{fossil}$ and $CO_2$-$e_{gen}$)

To determine carbon dioxide emissions displaced and non-carbon dioxide emissions generated for liquid biofuel (renewable diesel), it was necessary to first calculate the useful (available) energy. Assuming a renewable diesel production volume of 0.32 kL per tonne of dry wood [51,52], and a moisture content of 12% (wood shavings), the renewable diesel production of dry wood biomass was estimated to be 0.36 kL tonne$^{-1}$. Using the standardized value of 50% carbon content in wood, the available energy per tonne of carbon for liquid fuel was calculated with:

$$E_{avail(liquid)} = E_{content(biofuel)} \times \frac{V_{prod}}{C_{content}/100} \tag{8}$$

where $E_{content(biofuel)}$ is the energy content of one kilolitre of biofuel (GJ kL$^{-1}$), $V_{prod}$ is the volume of biofuel produced per tonne of dry wood biomass (kL t$_{biomass}$$^{-1}$), and $C_{content}$ is the carbon content (%), used to convert units from biomass to carbon.

Using the available energy figures and the NGA [47] emission factor for transport fuels of the substituted fossil fuel type (traditional diesel oil), carbon dioxide emissions avoided through combustion were calculated with:

$$CO_2 - e_{fossil(liquid)} = E_{avail(liquid)} \times \left( EF_{liquid}/1000 \right) \tag{9}$$

where $CO_2$-$e_{fossil(liquid)}$ is the carbon dioxide emissions (tCO$_2$-e tC$^{-1}$), $E_{avail(liquid)}$ is the useful (available) energy, and $EF_{liquid}$ is the emission factor for the production of liquid biofuel (kgCO$_2$-e GJ$^{-1}$).

Using the available energy figures and the NGA [47] emission factors for methane and nitrous oxide generated through combusting biofuels for transport energy purposes, the non-carbon dioxide emissions generated in the bioenergy type were calculated with:

$$CO_2 - e_{gen(liquid)} = E_{avail(liquid)} \times \frac{EF_{non-CO_2(liquid)}}{1000} \tag{10}$$

where $E_{avail(liquid)}$ is as above, and $EF_{non-CO_2(liquid)}$ is the non-carbon dioxide emission factor for liquid fuel (kgCO$_2$-e GJ$^{-1}$).

### 2.5.4. Transport Emissions ($CO_2$-$e_{transp}$)

To calculate carbon dioxide emissions associated with transportation, it was necessary to first calculate the carbon dioxide emissions generated per volume of transport fuel, assumed to be traditional diesel, using NGA [47] energy content and emission factors. Assuming an energy content of one kilolitre of diesel ($E_{content(diesel)}$) of 38.6 GJ kL$^{-1}$, fuel intensity ($Fuel_{intens}$) of 0.16 L tC$^{-1}$ km$^{-1}$ (assuming 0.07 L t$_{biomass}$ km$^{-1}$ [53], 15% moisture content, and a distance of 50 km for CHP and wood pellets and 300 km for renewable diesel, transport emissions were calculated as:

$$CO_2 - e_{transp(CHP,pellet)} = E_{content(diesel)} \times Fuel_{intens} \times 50 \tag{11}$$

$$CO_2 - e_{transp(liquid)} = E_{content(diesel)} \times Fuel_{intens} \times 300 \tag{12}$$

### 2.5.5. Annual Net GHG Emissions Avoided ($CO_2$-$e_{offset\_ha}$ and $CO_2$-$e_{offset\_total}$)

For each scenario, net GHG emissions avoided (offset) by substituting a traditional fossil fuel type with a bioenergy alternative were estimated on a per hectare per year basis with:

$$CO_2 - e_{offset\_ha} = C_{residue} \times CO_2 - e_{avoid} \qquad (13)$$

where $CO_2\text{-}e_{offset\_ha}$ is the GHG emissions avoided for each scenario (tCO$_2$-e ha$^{-1}$ year$^{-1}$), $C_{residue}$ is the carbon content in available residues estimated using FullCAM (tC ha$^{-1}$ year$^{-1}$), and $CO_2\text{-}e_{avoid}$ is the net GHG emissions avoided in the energy substitution scenario, expressed as carbon dioxide equivalents (tCO$_2$-e/tC) (Equation (1)). Conversion to total (regional) annual avoided (offset) potential was achieved by multiplying $CO_2\text{-}e_{offset\_ha}$ by the total area (*A*), in hectares:

$$CO_2 - e_{offset\_total} = CO_2 - e_{offset\_ha} \times A \qquad (14)$$

### 2.6. Uncertainty and Sensitivity Analysis

To investigate the variability of emission offsets based on uncertainties of inputs, a Monte Carlo analysis was performed on selected variables using Microsoft Excel software. The effect of energy conversion efficiencies for CHP and wood pellets, fuel intensity of renewable diesel, and variability in the proportion of stem available for extraction were investigated. For conversion efficiency, the impact of a ±5% variation from the 70% base rate for CHP, and 75% base rate for pellets, was investigated. For renewable diesel, the ±10% variation from the fuel intensity rate of 320 L per tonne of biomass was investigated. To account for uncertainty in the amount of residue left on-site, the proportion of stem and bark carbon available for extraction was varied over the range 95% ± 5% of the total reported by the FullCAM modelling. Random variates within the specified ranges were sampled from a rectangular distribution, and 1000 Monte Carlo iterations were run to calculate the average outcome and the variability (reported as the standard deviation). Covariances between random variables were assumed to be zero, reflecting the assumption that they are statistically independent.

The sensitivity of the results to the distance required to transport residue feedstock to a bioenergy processing facility was explored, with four distance scenarios (50 km, 100 km, 200 km, and 300 km), For CHP and pellets, the base case was 50 km, and 300 km for renewable diesel, given the distances from the study site to existing or proposed bioenergy facilities.

### 3. Results

### 3.1. Carbon Stocks

Table 3 shows the results of the FullCAM simulations for the different tree fractions, forest management treatments, and residue utilization alternatives.

**Table 3.** Carbon in forest harvest residue for two residue utilization alternatives and forest treatments. Values are the total residues produced over the course of a single 30-year rotation.

| Residue Alternative | Forest Treatment | Stems (tC ha$^{-1}$) | Branches (tC ha$^{-1}$) | Bark (tC ha$^{-1}$) | Total (tC ha$^{-1}$) |
|---|---|---|---|---|---|
| 1 | Thinning | 0.84 | 4.79 | 0.12 | 5.75 |
| | Final harvest | 3.14 | 17.97 | 0.42 | 21.53 |
| | Total | 3.98 | 22.76 | 0.54 | 27.28 |
| 2 | Thinning | 15.92 | 4.79 | 2.04 | 22.75 |
| | Final harvest | 3.14 | 17.97 | 0.42 | 21.53 |
| | Total | 19.06 | 22.76 | 2.46 | 44.28 |

Over a full rotation, 27.28 tC ha$^{-1}$ of forest residue biomass is predicted to be available for use under residue utilization alternative 1, and 44.28 tC ha$^{-1}$ under residue utilization alternative 2 (Table 3). Branches at final harvest comprised the largest residue component of any fraction, across all forest treatments (17.97 tC ha$^{-1}$). At thinning, stems had the largest accumulated carbon (15.92 tC ha$^{-1}$) in residue utilization alternative 2, where it

was assumed that 95% of thinned stems were available for bioenergy, compared with alternative 1 (0.84 tC-ha$^{-1}$), where just 5% of stems were assumed to be available. Stems at final harvest accounted for just a small proportion of carbon (3.14 tC ha$^{-1}$) available for bioenergy, which is consistent with expectations that this fraction provides the primary, merchantable product to mill. Carbon in bark accounted for a small proportion of the total carbon in both residue utilization alternatives (0.54 tC ha$^{-1}$ in alternative 1; 2.46 tC ha$^{-1}$ in alternative 2) available for bioenergy. These bark volumes ranged from 2% to 5.3% of the total residue material and included only bark on stems, not bark on branches.

Information on past harvesting activity for the case study area was used to generalize and scale the results to the whole plantation estate (case study site), which suggested an average harvesting rate of 3.3%, or 2833 hectares per year. Based on the simplifying assumption of continuous harvesting and replacement, with an even representation of coppicing across the estate, an average of 77,293 tC year$^{-1}$ in residues was expected to be available across the whole plantation (average of 0.91 tC ha$^{-1}$ year$^{-1}$) for residue utilization alternative 1; and 125,460 tC year$^{-1}$ (average of 1.48 tC ha$^{-1}$ year$^{-1}$) for residue utilization alternative 2.

### 3.2. Avoided GHG Emissions

Table 4 shows the carbon dioxide emissions associated with producing the equivalent energy to that available in the residue for the three different bioenergy types (or scenarios). Standard deviation values depict the effects of variation in the uncertainty analysis for energy conversion efficiencies (CHP and pellets), renewable diesel intensity, and thinned stem and bark utilization. The combustion carbon dioxide emissions represent the carbon dioxide emissions associated with combusting the traditional fossil fuel source, which are avoided in the proposed scenarios. Methane and nitrous oxide emissions (non-CO$_2$), and transport emissions generated by bioenergy, expressed as carbon dioxide equivalents, are also shown in Table 4.

**Table 4.** Carbon dioxide emissions associated with the combustion of fossil fuel type (avoided), GHG emissions associated with bioenergy (generated), and net GHG emissions avoided.

| Scenario [1] | Combustion Carbon Dioxide Emissions Avoided (t CO$_2$-e tC$^{-1}$) | Non-CO$_2$ GHG Emissions Generated by Bioenergy (t CO$_2$-e tC$^{-1}$) | Transport Carbon Dioxide Emissions (t CO$_2$-e tC$^{-1}$) [2] | Net GHG Emissions Avoided (tCO$_2$-e tC$^{-1}$) |
|---|---|---|---|---|
| 1 | 5.15 ± 0.30 | 0.031 ± 0.002 | 0.022 | 5.10 ± 0.30 |
| 2 | 1.44 ± 0.04 | 0.034 ± 0.001 | 0.022 | 1.38 ± 0.04 |
| 3 | 1.74 ± 0.20 | 0.062 ± 0.007 | 0.130 | 1.53 ± 0.19 |

[1] Scenario 1 (CHP from residue replacing grid electricity); scenario 2 (wood pellets from residue replacing natural gas); scenario 3 (renewable diesel from residue replacing diesel). [2] Transport emissions are reported for the base case for each scenario (50 km for scenarios 1 and 2, and 300 km for scenario 3).

In this case study, biomass-fed CHP that replaces coal-fired electricity had the highest mitigation potential, with 5.10 ± 0.30 tonnes of GHG emissions avoided for each tonne of biomass carbon combusted. This compares to 1.53 ± 0.19 and 1.38 ± 0.04 tonnes of avoided emissions for renewable diesel replacing traditional diesel and wood pellets replacing natural gas, respectively. Renewable diesel generated the most non-carbon dioxide emissions, 0.062 ± 0.007 tonnes of equivalent carbon dioxide emissions for each tonne of carbon available for energy generation, and the highest transport carbon dioxide emissions (0.13 t CO$_2$-e tC$^{-}$), given the longer travel distance to the proposed bioenergy facility in this case study.

### 3.3. Annual GHG Emissions and Sensitivity Analysis

The average GHG emissions per hectare potentially avoided each year for each scenario and residue utilization alternative are shown in Table 5. Additionally, average GHG emissions avoided are expressed on a whole plantation estate basis, representing the poten-

tial mitigation using the available residue for bioenergy each year from the case study area. The sensitivity of the results to change in transport distance (50 km, 100 km, 200 km, and 300 km) are also included in Table 5.

**Table 5.** Sensitivity analysis for the impact of transport distances on GHG emission offsets for each scenario for residue utilization alternatives 1 and 2 (average and standard deviation for a per hectare and site basis).

| Scenario | Residue Alternative | Average per ha GHG Emissions Avoided per Year ($\pm$s.d.) ($tCO_2$-e ha$^{-1}$ year$^{-1}$) | | | | Average Site GHG Emissions Avoided per Year ($\pm$s.d.) ($tCO_2$-e year$^{-1}$) | | | |
|---|---|---|---|---|---|---|---|---|---|
| | | 50 km | 100 km | 200 km | 300 km | 50 km | 100 km | 200 km | 300 km |
| 1 | 1 | **4.61 $\pm$ 0.27** | 4.60 $\pm$ 0.27 | 4.53 $\pm$ 0.27 | 4.51 $\pm$ 0.27 | **13,064 $\pm$ 763** | 12,993 $\pm$ 761 | 12,894 $\pm$ 767 | 12,773 $\pm$ 772 |
| | 2 | **7.36 $\pm$ 0.43** | 7.32 $\pm$ 0.43 | 7.24 $\pm$ 0.44 | 7.2 $\pm$ 0.44 | **20,798 $\pm$ 1228** | 20,681 $\pm$ 1242 | 20,595 $\pm$ 1230 | 20,443 $\pm$ 1228 |
| 2 | 1 | **1.24 $\pm$ 0.04** | 1.23 $\pm$ 0.04 | 1.19 $\pm$ 0.04 | 1.15 $\pm$ 0.04 | **3538 $\pm$ 105** | 3480 $\pm$ 103 | 3364 $\pm$ 108 | 3251 $\pm$ 106 |
| | 2 | **1.99 $\pm$ 0.07** | 1.96 $\pm$ 0.06 | 1.90 $\pm$ 0.06 | 1.84 $\pm$ 0.06 | **5653 $\pm$ 189** | 5548 $\pm$ 184 | 5380 $\pm$ 183 | 5194 $\pm$ 177 |
| 3 | 1 | 1.49 $\pm$ 0.18 | 1.47 $\pm$ 0.17 | 1.44 $\pm$ 0.17 | **1.38 $\pm$ 0.17** | 4203 $\pm$ 493 | 4186 $\pm$ 487 | 4088 $\pm$ 495 | **3929 $\pm$ 490** |
| | 2 | 2.38 $\pm$ 0.27 | 2.35 $\pm$ 0.28 | 2.29 $\pm$ 0.28 | **2.23 $\pm$ 0.29** | 6684 $\pm$ 781 | 6596 $\pm$ 810 | 6477 $\pm$ 789 | **6320 $\pm$ 782** |

Note: Base transport distances for each of the scenarios are shown in bold.

With residue utilization 1, there is the potential to avoid 4.61 tonnes of GHG emissions per hectare per year if CHP generated by harvest residue were to replace traditional fossil fuel electricity at a base transport distance of 50 km. This compares to 1.38 tonnes of emissions per hectare per year for renewable diesel at a base transport distance of 300 km, and 1.24 tonnes of emissions per hectare per year for wood pellets at a base transport distance of 50 km, when these renewable sources replace fossil fuel sources. As expected, the net GHG emissions avoided were higher for all scenarios under residue utilization alternative 2, where the larger harvested volumes reflect the inclusion of 95% of stems and bark at thinning (compared to 5% in alternative 1).

Based on the annual harvest residues available for bioenergy from the Toolara–Tuan Forest Estate, it would be possible to avoid (offset) 12,773 to 20,798 tonnes of GHG emissions per year by replacing coal-fired electricity with CHP (scenario 1). Wood pellets replacing natural gas could avoid 3251 to 5653 tonnes of emissions per year (scenario 2); replacing diesel could avoid 3929 to 6684 tonnes of emissions per year (scenario 3). For all scenarios, there was a negligible decrease in emission offsets with increased transport distances to the bioenergy facility (see Table 5).

## 4. Discussion

This study found that annual harvest residues from a managed softwood plantation in southeast Queensland can provide carbon mitigation benefits when used for bioenergy. The study results revealed that bioenergy replacing coal-fired electricity provides the largest mitigation opportunity, consistent with [23,24]. In Queensland, electricity is mostly generated by burning black coal, a carbon-intensive fuel source, that emits fewer emissions relative to brown coal (used widely for electricity generation in other parts of Australia), but more than other fossil fuels considered in the study: diesel and natural gas. This study suggests that potential GHG emissions offsets may be up to four times higher if forest residue bioenergy displaces coal-fired electricity compared with diesel or natural gas.

From a practical perspective, replacing coal-fired electricity with combined heat and power is a realistic short-term option, given the technological and operational capacity to upscale and adapt existing practices to increase forest residue for combined heat and power at the study site. Laminex Group Pty Ltd. (Maryborough, Australia), a wood-processing facility situated at the edge of the Toolara–Tuan Forest Estate, utilizes forest residues for heat and electricity (CHP) at its plant. Opportunities exist to expand use of residues, to include harvest residues, for CHP at both this facility and others co-located at the plantation. Additionally, there is future scope to sell excess energy to the State electricity.

Mitigation opportunities were comparable between diesel and natural gas when these were substituted for renewable diesel and wood pellets, respectively. Similarly to combined heat and power, there are existing or developing facilities with a capacity to utilize forest harvest residues for pellets or renewable diesel in Queensland. Co-located at the plantation site is a pellet production company, Altus Renewables Limited, with a capacity to produce 125,000 tonnes of wood pellets per year [45] and scope to expand further. International demand for pellets is expected grow, and there is likely to be greater demand in the domestic Australian market as States work towards increasing the share of renewables in the energy mix.

Renewable diesel may currently be a hypothetical bioenergy scenario for Queensland; however, Southern Oil Pty Ltd. (Gladstone, Australia) has established an advanced biofuels pilot project at the Northern Oil Refinery at Gladstone. Beginning with bagasse (sugarcane residues) and prickly acacia (invasive pest species), the plant has scope to further expand and utilize other woody feedstocks in the future [46]. Renewable diesels, replacing traditional fossil fuel-based diesels, represent one of the main, short-term opportunities to reduce transport emissions [9], which is particularly relevant to Queensland, as well as Australia more broadly, given the extensive road networks and heavy reliance on road freight.

Initiatives to promote sustainable and renewable energies and emissions reductions exist at both state and national level. Recently, Queensland has made specific commitments to biofuels and bioproducts [17]. With the skills, resources, and technology already existing in forestry and agriculture, along with biofuel blend mandates in the transport industry, Queensland is well positioned to develop a biofuels industry based on residue feedstocks [54]. Waste-to-bioenergy is well positioned to play a significant role in assisting Queensland to reach its 50% renewable energy by the 2030 target [55]. This study further supports the opportunity for forest woody residues to contribute to carbon mitigation in Queensland and Australia.

Under the Australian Government's Renewable Energy Target (RET) scheme, electricity providers are required to meet regulated renewable energy obligations [15]. The Emissions Reduction Fund (ERF) provides financial incentives for emission reduction technologies through tradeable carbon credits [16]. 'Wood waste' is specifically identified in the Australian Government's Renewable Energy (Electricity) Act 2000 as an eligible renewable energy source for industrial electricity projects attracting ERF carbon credits [56]. Similarly, fuel switching to sources such as renewable diesel that reduce emissions in land transportation and aviation are eligible for ERF carbon credits.

Carbon pricing fluctuates in response to many market-based and other forces, including supply and demand and political and planning processes. The current price (September 2021) of Australian carbon credit units (ACCUs) is AUD 26 (where each ACCU is equivalent to one tonne of carbon dioxide equivalent ($tCO_2$-e) avoided), up from about AUD 15 per ACCU ($tCO_2$-e) during most of 2020. Using the estimate of carbon emissions avoided in this study, where bioenergy replaces coal-fired electricity, and the more conservative estimate for harvest residue availability (residue utilization alternative 1), AUD 339,664 of carbon credits may be possible. This assumes that a price of AUD 26/t $CO_2$-e and 13,064 tonnes of carbon are avoided (offset), and that these avoided emissions meet the additionality test.

In this study, two alternatives for thinned stems and associated bark were analysed. The more conservative assessment included a minor proportion (5%) of these fractions for bioenergy and more closely represents the current practice at the study site. Removal of higher proportions assumes that bioenergy provides the optimal utilization over alternative uses. Pulp logs and round logs have uses in panel production, landscaping, and construction products; there are emerging opportunities for residues in biochar [57] and bioplastic production [58]. Commercial contracts and revenue streams associated with alternative uses of residue material, particularly pulp logs and round logs from thinning operations, may impact the economic viability and therefore emission mitigation potential

of harvest residues used for bioenergy. Emission offsets will also be impacted by alternative fates of residues such as burning and decay, not considered in this study; likewise, there are opportunities to combine bioenergy generation with carbon capture and storage (BECCS), which may even lead to negative emissions [59,60].

The bark content of residue feedstock for bioenergy must be managed to avoid negative impacts, including pollutants (nitrogen oxides and sulphur dioxide emissions), ash content, reduced calorific value and therefore combustion efficiency, and plant equipment damage. Bark in softwood plantations is not as easily removed in situ from stems as in hardwood plantations and is transported to processing facilities attached to the logs and branches. Given that the highest percentage of bark is found at the top of the stem [61], harvest residues, particularly those containing high proportions of stem tops and branches, are likely to have higher bark content than the 10% average by weight in pine logs [61]. At this site, bark currently contributes a higher-than-ideal proportion of the feedstock available for pellets [62], most likely due to bark included in the chipping of stems and branches. Although not currently in practice at the case study site, debarking equipment typically used at sawmills and other bioenergy facilities around the world could be employed to remove the bark if required [63]; however, this represents an additional financial cost.

This study did not consider costs associated with use of forest residue for bioenergy. Ref. [24] identified some of these costs to include feedstock costs, costs of extraction, and fluctuations in price of fossil fuels. Transportation and hauling biomass are widely recognized as the most significant forest bioenergy costs [23,31,64]. Where bioenergy facilities are co-located at the forest, or close by, there are likely to be lower costs, and therefore higher net emission offset credits. This study shows that although transport costs may be significant in bioenergy projects, transport emissions are minor relative to the overall emissions profile.

Concerns have been raised about the sustainability of forest bioenergy [65,66], in particular, the diversion of wood from native or unmanaged forests, high-quality stems from managed plantations, and land use changes that might lead to deforestation [67]. High energy demands can lead to increased prices and the diversion of woody biomass from other products [68]. When these diversions occur, there can be implications for the assumption of carbon neutrality, food security, and wood product industries. The case study site, Toolara–Tuan Forest Estate, is a well-established and sustainably managed plantation with long-term contracts for the supply of high-quality stems for timber products. Co-located at the site is a mill, which processes sawlogs for construction materials. Additionally co-located are processing plants utilizing residues for secondary products such as panelboards as well as bioenergy products for both on-site use and sale to the market. Given this and the very large difference in prices between bioenergy products and traditional forest products, there appears to be limited risk that forest biomass material, other than residues, will find their way to bioenergy at this regional site; it is unlikely to attract the same concerns as native forest residues, as reported by [33].

Additionally, consideration of the impacts of the removal of forest residue for bioenergy on soil health and nutrition is important. Soil health and nutrition can be supported by ensuring suitable volumes of nutrient-rich tree fractions remain on-site to return nutrients to the soil [28]. Pine needles are the most nutrient-dense fraction of softwoods [69]; in this study, needles were not considered as a bioenergy feedstock. As well as pine needles, this study excluded cones, and at least 5% of each of the other tree fractions (stems, branches, bark) from the residue utilization. Soil health and nutrition in managed plantations are also maintained through silviculture practices that include routine monitoring of sites and fertilizer application. Thus, harnessing increased volumes of harvest residue for bioenergy should be supported with the ongoing monitoring of soil health and nutrition.

## 5. Conclusions

This case study has presented the modelling of forest harvest residue availability in Queensland to examine bioenergy output and carbon impacts under a range of existing

and hypothetical scenarios. It employed a national carbon accounting model, FullCAM, for a project-level assessment of forest residue bioenergy. It is the first such use of the model and demonstrates further research opportunities to apply the model and approach to other finer-scale biomass assessments such as agricultural residues and native forest thinning. The results of this study demonstrate that combined heat and power generated with harvest residue may provide the best opportunity to avoid (offset) GHG emissions as an alternative to fossil fuels. Significantly, the results confirm that there are large volumes of harvest residues produced each year and that a range of bioenergy final products replacing a fossil fuel source can provide emissions savings. This study found that emission offsets are not sensitive to increased distances to transport feedstock to bioenergy plant. Regarding the sustainability of feedstock sources for bioenergy, and the carbon emission implications of their use, this research presents finer-scale spatial and temporal estimates compared to other studies, and an important preliminary assessment of harvest residue for bioenergy. As Queensland continues to invest in its renewable energy and bio-products future, this research contributes to the overall discussion of feasibility and opportunities for industry and governments.

The modelling framework developed here required making a number of simplifying assumptions. It was assumed that the entire plantation was stocked with the hybrid taxon, although there remained a mix of taxa. It was also assumed that harvest would occur at a regular and consistent rate over the next 30 years, although the plantation management plan is to continue to reduce rotation length into the future. Conversion pathways and technological feasibility, as well as considerations of supply and quality, were excluded the study, as were direct and indirect GHG emissions associated with collecting and storing residues.

To further enhance our understanding, future research that looks more broadly at other sustainability factors is required. This study focused on carbon mitigation and did not consider social and economic considerations such as regional development, capital investment requirements, transportation costs, and the quality of supply. Future research that considers mill residues, technological constraints, and opportunity costs of existing and future alternative uses of available residues, such as biochar and bioplastics, is required to better understand the full scope of the potential of forest residues for bioenergy in Queensland.

**Supplementary Materials:** The following are available online at https://www.mdpi.com/article/10.3390/f12111570/s1, Figure S1: FullCAM regimes and thinning/final harvest tree fraction allocations. Table S1: Carbon content in residue (FullCAM). File: FullCAM plot files.

**Author Contributions:** Conceptualization, L.C.G.; methodology, L.C.G. and S.H.R.; software, S.H.R.; validation, L.C.G., S.H.R. and F.A.X.; formal analysis, S.H.R.; investigation, L.C.G.; data curation, L.C.G.; writing—original draft preparation, L.C.G.; writing—review and editing, S.H.R. and F.A.X.; visualization, L.C.G., S.H.R. and F.A.X.; project administration, L.C.G. All authors have read and agreed to the published version of the manuscript.

**Funding:** This research received no external funding.

**Institutional Review Board Statement:** Not applicable.

**Informed Consent Statement:** Not applicable.

**Data Availability Statement:** Data are available upon request from the authors.

**Acknowledgments:** The authors acknowledge the technical and conceptual support provided by Mark Brown, Ian Last, David Lee, and Sam Van Holsbeeck.

**Conflicts of Interest:** The authors declare no conflict of interest.

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
