# Peer review of "Greenhouse Gas Emission Offsets of Forest Residues for Bioenergy in Queensland, Australia"

_forests, doi:10.3390/f12111570_

Round 1

Reviewer 1 Report

This manuscript is dealing with forest residues for bioenergy to reduce greenhouse gas emission, which is interesting and meaningful, and provides an important preliminary assessment of harvest residue for bioenergy. The result is clearly presented and the discussion is reasonable.

My only concern is on carbon accounting, why is FullCAM selected? Thus, my suggestion is that more information and comments on FullCAM is needed to convince the reader, and the information should be included in the introduction sector, because FullCAM modeling is important and used in estimates of carbon stocks and flows of available harvest residues in this manuscript. One more thing is on the use of reference number. For instance, in Line 113 “--- southern Australia was investigated by [32], and [33] assessed native forest---”, I don’t this usage is popular. The information in the corresponding references should be more specifically presented and to be easier  understood.

Author Response

Point 1: My only concern is on carbon accounting, why is FullCAM selected? Thus, my suggestion is that more information and comments on FullCAM is needed to convince the reader, and the information should be included in the introduction sector, because FullCAM modeling is important and used in estimates of carbon stocks and flows of available harvest residues in this manuscript.

Response 1: FullCAM is the national GHG accounting model used to track GHG emissions and report to international frameworks. It is also used by companies reporting emission savings for credits under the national emissions reduction fund

·         Added a summary to the Introduction to explain the choice of FullCAM in this project. Has lead to some minor re-organisation of sources 41-44.

Point 2: One more thing is on the use of reference number. For instance, in Line 113 “--- southern Australia was investigated by [32], and [33] assessed native forest---”, I don’t this usage is popular. The information in the corresponding references should be more specifically presented and to be easier  understood.

Response 2: Edited the style of writing, specifically the reference to sources in the Introduction for lines 108-124.

·         Introduction improved and strengthened with these changes – as requested by reviewers

Reviewer 2 Report

The article is well written, has current references and follows a clear line of research. Although it is an average paper, it provides scenario analysis which makes it interesting. Reading it was easy, however it takes some minor corrections:
1. What was the intensity of the sampling carried out?
2. The estimate of the biomass determined at each site should appear and not only the carbon, that would give us a clearer idea of the biomass, limiting only to biomass to carbon conversion coefficients limits the reader in sizing the study.
3. Did you use a statistical program for the tests? that is not clear. Monte Carlo analysis is used, but they don't mention in which package they used it: R, SAS, Matlab?

Author Response

Point 1: What was the intensity of the sampling carried out?

Response 1: As this was a desktop based assessment no actual sampling was carried out.

Point 2: The estimate of the biomass determined at each site should appear and not only the carbon, that would give us a clearer idea of the biomass, limiting only to biomass to carbon conversion coefficients limits the reader in sizing the study.

Response 2: As Carbon is assumed to account for half of dry biomass weight, biomass values can be derived by simply multiplying the carbon values in Table 3 by two. This is described in Table 2 and in line 330 etc. Biomass values were not separately included to ensure results were simply and clearly displayed.  

Point 3: Did you use a statistical program for the tests? that is not clear. Monte Carlo analysis is used, but they don't mention in which package they used it: R, SAS, Matlab?

Response 3: Yes, have added details of the software package (MS Excel) at line 388-9

Reviewer 3 Report

Dear Authors, 

As per my view manuscript is written well by taking care of novelties and scientific way of writing.  Present paper is interesting from carbon mitigation point of view and could be helpful for the regional and global researchers. The present paper need minor revision before acceptable for publication, kindly see the comments below to improve the quality of the manuscript. 

Kindly write GHG or GHGs thoroughly, plz

Figure 1 need to be prepared using RS-GIS, showing latitute and longitude 

Line no 588-591: text need support of past literature (see Land Degradation and Development, 2021, https://doi.org/10.1002/ldr.3984;

CATENA, 207, 2021, https://doi.org/10.1016/j.catena.2021.105667)

Conclusion need to be summarised in 2-3 paragraph maximum, request authors to re-write in fruitful way by avoiding general statement 

Author Response

Point 1: Kindly write GHG or GHGs thoroughly, plz

Response 1: I have changed 'GHGs' at line 31 to GHG emissions to address this. All other references in the text are 'GHG'.

Point 2: Figure 1 need to be prepared using RS-GIS, showing latitute and longitude 

Response 2: Latitude and longitude lines have been added to the Figure in the updated file. The authors feel that, on the original figure, the map inset provides the location area in the context of the Queensland/Australia and the precise latitude/longitude (updated at line 143 to reflect the central Forest office location) of the estate has been provided in the main text, and that latitude/longitude on the map may unnecessarily distract. If preferring to include these, the revised file now contains these lines.

Point 3: Line no 588-591: text need support of past literature (see Land Degradation and Development, 2021, https://doi.org/10.1002/ldr.3984;

  • CATENA, 207, 2021, https://doi.org/10.1016/j.catena.2021.105667)

Response 3: Authors have chosen not to cite these at this time.

Point 4: Conclusion need to be summarised in 2-3 paragraph maximum, request authors to re-write in fruitful way by avoiding general statement

Response 4: Conclusion has been revised and reduced to be more specific and targeted to the project. It is now 3 paragraphs in length.